# Patient-level and practice-level factors associated with consultation duration: a cross-sectional analysis of over one million consultations in English primary care

Sarah Stevens,[1] Clare Bankhead,[1] Toqir Mukhtar,[1] Rafael Perera-Salazar,[1] Tim A Holt,[1] Chris Salisbury,[2] F D Richard Hobbs,[1] on behalf of the NIHR School for Primary Care Research, Nuffield Department of Primary Care Health Sciences, University of Oxford

[1]Nuffield Department of Primary Care Health Sciences, University of Oxford, Oxford, UK
[2]Centre for Academic Primary Care, School of Social and Community Medicine, University of Bristol, Bristol, UK

**Correspondence to**
Sarah Stevens;
sarah.stevens@phc.ox.ac.uk

## ABSTRACT

**Objectives** Consultation duration has previously been shown to be associated with patient, practitioner and practice characteristics. However, previous studies were conducted outside the UK, considered only small numbers of general practitioner (GP) consultations or focused primarily on practitioner-level characteristics. We aimed to determine the patient-level and practice-level factors associated with duration of GP and nurse consultations in UK primary care.

**Design and setting** Cross-sectional data were obtained from English general practices contributing to the Clinical Practice Research Datalink (CPRD) linked to data on patient deprivation and practice staffing, rurality and Quality and Outcomes Framework (QOF) achievement.

**Participants** 218 304 patients, from 316 English general practices, consulting from 1 April 2013 to 31 March 2014.

**Analysis** Multilevel mixed-effects models described the association between consultation duration and patient-level and practice-level factors (patient age, gender, smoking status, ethnic group, deprivation and practice rurality, number of full-time equivalent GPs/nurses, list size, consultation rate, quintile of overall QOF achievement and training status).

**Results** Mean duration of face-to-face GP consultations was 9.24 min and 5.32 min for telephone consultations. Nurse face-to-face and telephone consultations lasted 9.70 and 5.73 min on average, respectively. Longer GP consultation duration was associated with female patient gender, practice training status and older patient age. Shorter duration was associated with higher deprivation and consultation rate. Longer nurse consultation duration was associated with male patient gender, older patient age and ever smoking; and shorter duration with higher consultation rate. Observed differences in duration were small (eg, GP consultations with female patients compared with male patients were 8 s longer on average).

**Conclusions** Small observed differences in consultation duration indicate that patients are treated similarly regardless of background. Increased consultation duration may be beneficial for older or comorbid patients, but the

## Strengths and limitations of this study

► This is a large-scale analysis of over one million consultations, using data known to be representative of the UK population.
► We have considered factors associated with the duration of both general practitioner (GP) and nurse consultations allowing comparison between the two.
► Appointment duration may be recorded with some error, but average durations were consistent with 10 min appointment slots.
► We were unable to examine how GP/nurse characteristics are associated with consultation duration, and this requires further study.

benefits and costs of increased consultation duration require further study.

## INTRODUCTION

Patient-facing general practice workload in England has increased by 16% since 2007.[1] This reflects an increase in both the rate and duration of consultations. Consultation duration may be influenced by patient, practitioner and practice-level characteristics. At the practice level, previous studies have shown that shorter consultation duration is associated with greater practice list size[2] and workload.[3] The influence of practice rurality (rural compared with urban) is unclear with some studies indicating that rurality is negatively associated with consultation duration[2 4] and others demonstrating a positive association.[5] Relevant practitioner characteristics associated with longer consultations include female gender,[6] older age,[3 5] but conversely, lesser experience.[6]

Finally, longer consultations have been shown to be associated with patient characteristics, including female gender,[3–5 7] older age,[2–5 7] greater number of presenting problems[3–5 7 8] and higher level of education[3] or socio-economic status.[5]

However, many previous studies have been conducted in countries other than the UK, and findings may not be generalisable to the National Health Service (NHS).[2–5] Studies within the UK provide limited up-to-date evidence having been conducted some time ago using data on a relatively small number of consultations[7] or having focused on practitioner-level characteristics alone.[6] Although a 2013 paper studied the association between practice, practitioner and patient-level characteristics and the number of presenting problems, demonstrating that the number of presenting problems is also associated with consultation duration, direct links between patient and practice characteristics and duration were not studied.[8] Finally, previous work has considered duration of general practitioner (GP) consultations only, despite nurse consultations accounting for approximately one quarter of the overall UK primary care consultation rate in 2013/2014.[1] Hence, we aimed to determine the patient and practice characteristics associated with increased duration of GP and nurse consultations in UK primary care in contemporary data.

## METHODS

Consultation and patient data were obtained from the Clinical Practice Research Datalink (CPRD), a research database of anonymised patient records drawn from over 600 UK general practices.[9] English practices consenting to CPRD's data linkage scheme were included in the study if they contributed data covering any part of the study period (1 April 2013 to 31 March 2014) and were defined as 'up-to-standard' (CPRD definition of continuous high-quality data recording fit for use in research). All non-temporary patients registered at eligible practices for at least 1 day during the study period were included. Due to data volume, analysis was limited to a 10% simple random sample from each age–sex strata of eligible patents and those who consulted at least once during the study period.

CPRD data were linked to practice data on staffing,[10] rurality,[11] Quality and Outcomes Framework (QOF) performance measures[12] and patient Index of Multiple Deprivation (IMD). IMD data were supplied in quintiles by CPRD, who link patient postcodes to publically available IMD scores and group data into quintiles at the English national level. Staffing, rurality and QOF data were downloaded from NHS digital (formerly the Health and Social Care Information Centre), and continuous variables were grouped prior to linkage with CPRD data. This was a requirement of the Independent Scientific Advisory Committee to CPRD to limit the possibility of identifying individual CPRD practices. The approved protocol (no 15_120R) is available from the authors.

Consultations in CPRD represent occasions on which a patient's electronic health record is opened. We analysed consultations that were identified as face-to-face or telephone consultations based on the variable 'consultation type', and those with a GP or nurse only, as indicated by the variable 'staff role'. We excluded consultations where the patient record was opened purely for administrative purposes by GPs, nurses or administrative staff (eg, to record test results) and home visit consultations (since recorded duration may merely represent the time taken to record the consultation after it has ended).

Mean consultation duration across practices was examined using histograms. Practices were grouped according to their average consultation duration (<5 and ≥5 min;<8 and ≥8 min;<10 and ≥10 min;<12 and ≥12 min and <15 and ≥15 min) and differences in their characteristics described.

Multilevel mixed-effects models were used to model the association between patient and practice characteristics and duration of GP or nurse consultations separately. Patient factors included as fixed-effects were age, gender, smoking status (current, former and never), ethnic group and quintile of IMD. Fixed-effects practice-level factors included were rurality, number of full-time equivalent (FTE) GPs, number of FTE nurses, list size (centred), rate of GP consultation (centred), rate of nurse consultation (centred), quintile of overall QOF achievement and practice training status (yes or no). Indicators for the patient and practice were included as random effects. All variables were entered into the models simultaneously and subsequently excluded in a stepwise fashion based on Z tests (binary and continuous variables) or $\chi^2$ tests (categorical variables) at the 5% level. Missing smoking status and ethnic group data were included as separate categories in the models.

## RESULTS

In total, 3 049 320 patients were eligible during the study period, of which 304 937 were randomly selected for inclusion. Of these, 218 304 consulting patients from 316 practices were included. The characteristics of the included patients and practices are given in table 1 and table 2, respectively. During the study period, 964 148 consultations were conducted by a GP, and 347 657 were conducted by a nurse. The majority of consultations (1 155 040; 88%) were face-to-face consultations. Mean duration of face-to-face GP consultations was 9.24 (SD=8.06) min compared with 5.32 (6.21) min for telephone consultations. Nurse consultations were longer, on average, than those with GPs; face-to-face and telephone nurse consultations lasted 9.70 (9.21) and 5.73 (6.29) min. A minority of practices conducted substantially shorter or longer consultations on average (online supplementary figure S1).

**Table 1** Characteristics of included patients (N=218 304)

| | Mean/n | SD/% |
|---|---|---|
| Female gender | 121 107 | 55.5 |
| Age group (years) | | |
| 0–14 | 36 371 | 16.7 |
| 15–24 | 23 020 | 10.5 |
| 25–44 | 55 316 | 25.3 |
| 45–64 | 57 000 | 26.1 |
| 65–74 | 24 086 | 11.0 |
| 75+ | 22 511 | 10.3 |
| Smoking status | | |
| Non-smoker | 82 327 | 37.7 |
| Current smoker | 37 286 | 17.1 |
| Ex-smoker | 40 834 | 18.7 |
| Unknown | 57 857 | 26.5 |
| IMD | | |
| First quintile (least deprived) | 48 363 | 22.2 |
| Second quintile | 47 948 | 22.0 |
| Third quintile | 41 825 | 19.2 |
| Fourth quintile | 41 953 | 19.2 |
| Fifth quintile (most deprived) | 34 750 | 15.9 |
| Unknown | 3465 | 1.6 |
| Ethnic group | | |
| White | 118 063 | 54.1 |
| Asian | 6008 | 2.8 |
| Chinese | 491 | 0.2 |
| Black | 3908 | 1.8 |
| Mixed/other | 4374 | 2.0 |
| Unknown | 85 460 | 39.2 |

IMD, index of multiple deprivation.

**Table 2** Characteristics of included practices (N=316)

| | Mean/n | SD/% |
|---|---|---|
| List size | 9649.7 | 4648.4 |
| Training practice | | |
| Yes | 126 | 39.9 |
| Unknown | 2 | 0.6 |
| Rurality | | |
| Not rural (urban >10 000—less sparse) | 267 | 84.5 |
| Rural (hamlet/village/town and fringe) | 49 | 15.5 |
| GP consultation rate (per 10 000 person-years) | 37 441.0 | 13 043.6 |
| Nurse consultation rate (per 10 000 person-years) | 13 217.3 | 7580.5 |
| No of FTE GPs | | |
| ≤2 | 44 | 13.9 |
| >2 and ≤4 | 74 | 23.4 |
| >4 and ≤6 | 101 | 32.0 |
| >6 and ≤8 | 55 | 17.4 |
| >8 and ≤19 | 40 | 12.7 |
| Unknown | 2 | 0.6 |
| No of FTE nurses | | |
| ≤2 | 188 | 59.5 |
| >2 and ≤4 | 65 | 20.6 |
| >4 and ≤6 | 20 | 6.3 |
| >6 and ≤8 | 6 | 1.9 |
| >8 and ≤19 | 4 | 1.1 |
| Unknown | 33 | 10.4 |
| QOF performance | | |
| First quintile (poorest performance) | 50 | 15.8 |
| Second quintile | 49 | 15.5 |
| Third quintile | 59 | 18.7 |
| Fourth quintile | 82 | 26.0 |
| Fifth quintile (best performance) | 73 | 23.1 |
| Unknown | 3 | 1.0 |

FTE, full-time equivalent; GP, general practitioner; QOF, quality and outcomes framework.

## GP consultations

Practice characteristics by average length of GP consultation are described in online supplementary table S1. Practices conducting longer consultations had a lower rate of GP consultation, but the relationship between other characteristics was less clear. Full-model results for duration of GP consultations are given in online supplementary table S2, and variables were excluded in the following order: rate of nurse consultation (P=0.658), rurality (P=0.295), QOF performance (P=0.204), FTE nurses (P=0.063), FTE GPs (P=0.115) and list size (P=0.552). This yielded the final model in table 3. Female patients' GP consultations were 8.3 s longer on average, and patients aged 0–14 years had the shortest consultations. Those aged 45–64 years had the longest consultations; consultations were 1.5 min longer, on average, than consultations in 0 to 14 years. Although both ethnic group (P<0.001) and smoking status (P<0.001) were retained in the model, only the unknown categories showed significant associations: consultations with patients of unknown ethnicity were 11 s shorter than

those with White patients and consultations with patients of unknown smoking status were 19 s longer than those with non-smokers.

Duration of consultation decreased with increasing deprivation; consultations with patients in the most deprived quintile lasted 5 s less on average than consultations with the least deprived patients. Consultations in training practices were 44 s longer than those in practices that did not have trainee GPs, and telephone consultations were, on average, 5 min shorter than face-to-face consultations. Finally, for every 10% increase in consultation rate

**Table 3** Factors associated with duration of GP consultations

| | Change in duration (s) | P value | 95% CI |
|---|---|---|---|
| Female gender (male=reference) | 8.29 | 0.000 | 6.03 to 10.55 |
| Ethnic group (white=reference) | | | |
| Asian | 4.06 | 0.237 | −2.67 to 10.78 |
| Chinese | −6.40 | 0.603 | −30.49 to 17.69 |
| Black | −5.70 | 0.200 | −14.40 to 3.01 |
| Mixed/other | 4.30 | 0.289 | −3.64 to 12.24 |
| Unknown | −11.01 | 0.000 | −13.51 to −8.50 |
| IMD (first quintile=reference) | | | |
| Second quintile | 1.14 | 0.537 | −2.48 to 4.76 |
| Third quintile | −2.41 | 0.230 | −6.35 to 1.53 |
| Fourth quintile | −3.64 | 0.089 | −7.83 to 0.56 |
| Fifth quintile (most deprived) | −5.11 | 0.034 | −9.84 to −0.37 |
| Unknown | −11.36 | 0.058 | −23.12 to 0.39 |
| Smoking status (non-smoker=reference) | | | |
| Current smoker | −2.36 | 0.147 | −5.54 to 0.83 |
| Ex-smoker | 0.11 | 0.943 | −2.91 to 3.13 |
| Unknown | 18.65 | 0.000 | 14.56 to 22.73 |
| Age group (0–14 years=reference) | | | |
| 15–24 years | 55.70 | 0.000 | 50.11 to 61.29 |
| 25–44 years | 83.66 | 0.000 | 78.70 to 88.63 |
| 45–64 years | 89.81 | 0.000 | 84.75 to 94.87 |
| 65–74 years | 65.82 | 0.000 | 60.23 to 71.40 |
| 75+ years | 58.43 | 0.000 | 52.94 to 63.92 |
| Telephone consultation (face-to-face=reference) | −308.71 | 0.000 | −311.65 to −305.77 |
| Training practice (no=reference) | | | |
| Yes | 44.33 | 0.000 | 19.87 to 68.78 |
| Unknown | 121.58 | 0.148 | −43.02 to 286.17 |
| GP consultation rate (centred, per 1000 per 10 000 person-years) | −3.31 | 0.000 | −4.24 to −2.37 |
| Mean duration | 472.42 | 0.000 | 455.41 to 489.43 |

GP, general practitioner; IMD, index of multiple deprivation.

(1000 per 10 000 person-years), GP consultation duration decreased by 3 s.

In post hoc sensitivity analysis, we explored whether the association of duration with practice training status may be driven by consultations with trainee GPs alone, by adding a variable into the final model to indicate whether the GP conducting the consultation was a registrar or not. We found that consultations were on average 245 s longer with a GP registrar than otherwise, and practice training status became non-significant (P=0.0656, online supplementary table S3).

**Nurse consultations**

Practice characteristics, by average length of nurse consultation, are described in online supplementary table S4. Similarly to GP consultations, practices conducting longer consultations had a lower rate of nurse consultation. Full-model results for duration of nurse consultations are given in online supplementary table S5. Variables were removed from the full-model as follows: ethnic group (P=0.838), QOF performance (P=0.767), training practice (P=0.544), rurality (P=0.522), rate of GP consultation (P=0.547), FTE nurses (P=0.284) and list size (P=0.250).

In the final model (table 4), consultations with a nurse were 11 s shorter for women than for men. All age groups had longer consultations than those aged 0–14 years (up to maximum of 2 min longer in those aged 45–64 years). Current smokers and ex-smokers had longer nurse consultations than non-smokers, by an average of 27 and 15 s, respectively. Those in the second quintile of deprivation had longer consultations than those in the

**Table 4** Factors associated with duration of nurse consultations

| | Change in duration (s) | P value | 95% CI |
|---|---|---|---|
| Female gender (male=reference) | −11.06 | 0.000 | −15.24 to −6.88 |
| IMD (first quintile=reference) | | | |
| Second quintile | 7.58 | 0.024 | 1.01 to 14.16 |
| Third quintile | −0.40 | 0.912 | −7.56 to 6.75 |
| Fourth quintile | 5.49 | 0.158 | −2.14 to 13.11 |
| Fifth quintile (most deprived) | 8.04 | 0.066 | −0.53 to 16.62 |
| Unknown | −26.93 | 0.007 | −46.39 to −7.46 |
| Smoking status (non-smoker=reference) | | | |
| Current smoker | 26.67 | 0.000 | 20.80 to 32.55 |
| Ex-smoker | 15.20 | 0.000 | 9.80 to 20.60 |
| Unknown | 21.06 | 0.000 | 13.17 to 28.94 |
| Age group (0–14 years=reference) | | | |
| 15–24 years | 52.30 | 0.000 | 41.70 to 62.91 |
| 25–44 years | 71.85 | 0.000 | 62.54 to 81.16 |
| 45–64 years | 113.15 | 0.000 | 103.70 to 122.59 |
| 65–74 years | 73.81 | 0.000 | 63.71 to 83.90 |
| 75+ years | 75.68 | 0.000 | 65.61 to 85.75 |
| Telephone consultation (face-to-face=reference) | −279.34 | 0.000 | −288.18 to −270.50 |
| Number of FTE GPs (≤2=reference) | | | |
| >2 and ≤4 | −85.82 | 0.004 | −144.44 to −27.20 |
| >4 and ≤6 | −82.57 | 0.004 | −138.53 to −26.61 |
| >6 and ≤8 | −82.90 | 0.008 | −144.25 to −21.55 |
| >8 and ≤19 | −78.14 | 0.020 | −143.96 to −12.32 |
| Unknown | −235.93 | 0.140 | −549.15 to 77.29 |
| Nurse consultation rate (centred, per 1000 per 10 000 person-years) | −9.19 | 0.000 | −11.53 to −6.84 |
| Mean duration | 598.72 | 0.000 | 549.66 to 647.78 |

FTE, full-time equivalent; GP, general practitioner; IMD, index of multiple deprivation.

least deprived quintile, but there was no clear relationship in other groups. Those with unknown deprivation had shorter consultations. In practices with more than two FTE GPs, nurse consultations were between 78 and 86 s shorter, although the effect of FTE GPs was marginally significant (P=0.046) and was not significant when including list size in the model (P=0.109). Practices with a higher rate of nurse consultation had shorter consultations by an average of 9 s for every 10% increase in consultation rate (1000 consultations per 10 000 person-years).

## DISCUSSION

We have shown that duration of consultation is associated with both patient-level and practice-level characteristics. Increasing patient age is associated with increased consultation duration. Female patient gender increases the length of GP consultations and decreases the length of nurse consultations, and duration of nurse consultations

is increased in current and ex-smokers. GP consultations are longer in practices involved in GP training and with less deprived patients, but shorter in practices with a higher consultation rate. Although there is some variation in mean duration across practices, this is not explained by many of the practice characteristics studied.

## Strengths and limitations

This is a large-scale analysis of over one million consultations across England and therefore provides reliable estimates of association. Moreover, CPRD is broadly representative of the UK population,[9] and our results are likely to be representative of those consulting across England. A further strength is our separate consideration of GP and nurse consultations, allowing us to describe factors associated with the length of nurse consultations for the first time. A limitation is the consideration of general practice consultations only, and our results may not be generalisable to other settings (eg, walk-in centres).

Consultation duration in CPRD reflects the length of time a patient record is open within the practice computer system, recorded in whole minutes. There were instances in the data of very long (>60 min, 0.3%) and very short (apparent 0 min, 8.3%) consultations which we rounded to 60 min and 0.5 min, respectively. Long consultations may occur for genuine clinical need, but also if a staff member forgets to close a record. Apparent short consultations may occur if a record is opened incorrectly, if details of a straightforward consultation are entered only at the end of a consultation or if the type of consultation (eg, administrative) was miscoded. However, average durations were in line with a standard 10 min appointment window, and final model estimates were similar when excluding these extreme durations or including them without rounding (data not shown).

Due to missing data, we included some 'unknown' categories in our models. Previous research has shown that former smoking is under-reported in CPRD compared with UK national survey data,[13] so those with unknown status in this study may be more likely to be former smokers. Ethnic group data were drawn from hospital episodes data, so those with missing data may be healthier and consult less often (ethnic group was missing in 39% of patients, but only in 29% of consultations). Hence consultations in these patients may have been shorter and less complex. Ethnicity data are similarly poorly reported in CPRD[9]; hence, more detailed data are required to fully explore these associations.

We did not have data on GP and nurse characteristics, so were unable to examine their association with duration. Previous research outside of the UK has shown that consultations with older GPs are longer,[3 5] but UK-based research indicates that consultations are longer in those with lesser experience.[6] Our results regarding the association of duration with practice training status and GP registrar status are consistent with the UK research. However, we found that in practices which were not identified as training practices in the national data, 4.9% of GP consultations appeared to be conducted by GP registrars (compared with 11.6% in training practices). This indicates inaccuracies in coding either of staff role or of training practice status, and hence this finding needs further replication in future studies.

We did not examine the relationship between consultation duration and the number of presenting problems. A 2010 study indicated that GP consultation duration may be increased by 2 min for each additional presenting problem.[8] A similar large-scale analysis using CPRD presents many methodological difficulties and is the subject of ongoing work by the study authors.

### Comparison with the literature

Our contemporary results confirm previous research findings that increasing duration of GP consultation is associated with older patient age,[2–5 7] female patient gender[3–7] and socioeconomic status.[5] Older patient age and current or prior smoking are also associated with increased duration of nurse consultations. Although female patients have longer GP consultations, they have shorter nurse consultations, and the reason for this is unclear. Nurses may conduct more relatively straightforward consultations with women (eg, contraception reviews) compared with consultations with men.

Consultations are shorter in practices with a greater corresponding consultation rate, perhaps indicating that appointment lengths are limited to meet consultation demand, with little spare capacity in schedules. This is consistent with our previous work[1] showing that GP workload has increased by 16% in England since 2007.[1] Conversely, more problems may be dealt with in a longer consultation, reducing the need for repeat consults. This was previously demonstrated by a study in two practices where increased initial consultation duration was associated with a lower consultation rate in the following 4 weeks.[14]

### Implications

We observed small absolute differences in consultation duration, despite statistical significance for some factors. This may suggest that all patients are treated similarly and that consultation duration is equitable and in line with patient need. For example, we observed large differences related to patient age, which is likely to be confounded with comorbidity and complexity of consultation. However, our findings that more deprived patients have shorter consultations on average could indicate inequalities based on clinical need since more deprived patients have higher rates of premature mortality.[15] Practices with an older or comorbid patient list could increase the length of scheduled appointments to better match the required consultation time in these patients. Practices could also allow patients to choose their consultation length. This has been shown to improve doctor and patient experience, and patients could be educated to estimate their required time.[16]

Patients in the 31.7% of practices offering consultations less than 8 min long may receive significantly less GP care compared with those in the 31% of practices providing consultations of 10 or more minutes long, particularly when considering this difference across multiple appointments. However, we observed a small decrease in duration of 3 s for every 10% increase in consultation rate indicating a degree of trade-off between consultation length and number. The importance of consultation duration partly depends on its association with outcomes. Increasing duration has been shown to increase patient enablement and decrease GP stress.[17] A previous review suggested that doctors conducting longer consultations are more likely to offer health promotion advice and deal with long-term problems.[18] Longer consultations may also reduce prescribing rates[18] and be associated with more appropriate prescribing.[19] However, there is little strong evidence that duration is associated with patient satisfaction generally[18 20] or when GPs are preselected for poor communication,[21] although it may be associated with more patient-centredness.[6]

Evidence from exploratory trials suggests that increasing consultation duration (as part of a wider complex intervention) is highly cost-effective.[22] However, a recent review indicated that many studies assessing interventions to alter consultation duration are at high risk of bias; the effect of altering duration on the number of referrals, prescriptions or patient satisfaction is uncertain.[23] Further research is required to establish the benefits and costs of increasing consultation duration alone.

GP consultations are longer on average in practices hosting trainees. This may have implications for the future of general practice since GP recruitment has not kept pace with growth in the consulting population, and fewer trainees intend to stay in full-time clinical work.[24] Policy-makers and those responsible for recruitment should consider how the increased time required to train GPs can be accommodated given increasing workload pressures.[1]

**Contributors** FDRH and CS conceived the research, obtained funding and are joint principal investigators. FDRH, CB and CS drafted the protocol, which SS, TM, RP-S and TAH then contributed to. SS and TM were responsible for data management. SS did the statistical analyses and drafted the report, which FDRH, CS, CB, RP-S, TM and TAH then contributed to. SS is the guarantor and corresponding author.

**Funding** This project is funded by the National Institute for Health Research School for Primary Care Research (NIHR SPCR). FDRH is partly supported as an NIHR Senior Investigator, Director of the NIHR SPCR, Director of the NIHR CLARHC Oxford, Theme Leader of the NIHR Oxford BRC, NIHR Oxford DEC and Professorial Fellow of Harris Manchester College.

**Competing interests** SS, CS and RP-S report grants from the National Institute for Health Research School for Primary Care Research during the conduct of the study.

**Ethics approval** CPRD ISAC.

**Provenance and peer review** Not commissioned; externally peer reviewed.

**Data sharing statement** Data are available from CPRD directly: https://www.cprd.com/intro.asp.

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
