## [Reviewer comments · BMJ Open]

ARTICLE DETAILS

TITLE (PROVISIONAL)	Patient and practice level factors associated with consultation duration: a cross-sectional analysis of over 1 million consultations in English primary care.
AUTHORS	Stevens, Sarah; Bankhead, Clare; Mukhtar, Toqir; Perera, Rafael; Holt, Tim; Salisbury, Chris; Hobbs, Richard

VERSION 1 – REVIEW

REVIEWER	Denis Pereira Gray St Leonard's Research Practice, Exeter, UK
REVIEW RETURNED	19-Jul-2017

GENERAL COMMENTS	Thank you for inviting me to assess this article which has been submitted for publication. I am pleased to do so. I know several of the authors, including three of the professors, but I have no professional or financial links with any of the authors. I do not consider that I have any conflict of interest in commenting for you. STRENGTHS This article has many strengths: • It has been written by a strong academic team from a leading British Department of General Practice/Primary Care.• More than that, it comes from the School of Primary Care which links leading university departments of this kind.• The authors have access to one of the biggest and best known general practice data bases in the UK or indeed in Europe.• The methods used are appropriate for the data.• Some of the results reported are new..• One new finding is that for every 10% increase in the consultation rate, consultation duration decreased by 3 seconds, although the authors do not discuss if this is clinically important . When I started reading I had a keen sense of interest and expectation, but when I had finished reading, I felt disappointed. This article is worthy but dull. The authors have not really thought enough about the potential of their data and have simply run standard analyses in standard ways. They can do better than this. DISAPPOINTMENTS Some findings previously published • Many of their findings are already known. Indeed three of the authors (Hobbs et al., 2016) have already reported (in The Lancet) the mean length of GP consultations at 9.4 minutes.
---

- It has long been known that consultations with female patients are longer, although these authors show the gap is smaller than others only 8 seconds
Range/dispersion of average duration of consultations between practices
 - In an article entitled duration of consultations one would expect a critical analysis of the variation in duration of consultations between different general practices in which patients on average get longer or shorter consultations.
 - The whole pattern of NHS general practice since 1948 has been a steady lengthening of consultations. Although 10 minutes is the commonest unit of time, there have been calls for 15-minute booking (Irving et al., 2012)
 - In the NHS general practice from which I am writing this review, I conducted routine surgery appointments booked at a duration of 12 minutes (five to the hour), 20 years ago. This Practice has provided consultations of average duration of 15-16 minutes for several years and there are other NHS practices in Exeter offering 15-minute appointments routinely.
 - Longer consultations, particularly 15-minute consultations are now provided routinely in an important minority of general practices. The absence of any quantification of this important development is disappointing. Leading-edge general practice is particularly important and this article ignores it.
 - I think these authors, with their massive database, should report on the range between general practices how many practices consult at an average of more than 10 minutes, what number and what percentage? This is particularly important for general practitioners or general practices providing an average of 15 or more minutes.
 - The authors should, as they do in the article, report on the characteristics of such practices by training status, geography, practitioner characteristics etc.
 - Similarly, the authors should report on how many general practices offer shorter than average ie 7.5 minutes or of 5-minute appointments and the associated characteristics of these practices. Extreme durations
 - It is stated that “extreme durations” of all consultations studied which the authors dismiss (page 9) numbered 8.6% but this is more than one in 12 and so on average one per average surgery session.
 - This seems high and is higher than figures found in general practices which monitor this. Some more comment would be helpful.
- Patient satisfaction
- On page 10 it is stated “that there is no strong evidence that consultation length is associated with patient satisfaction” and two supporting references (16, 17) are given. This is a rather dogmatic statement, which I personally do not believe to be true. One of these references (Elmore et al., 2016) in my opinion is invalid. This was study of GPs selected as having poor communication skills, so it is not possible to draw conclusions about the effect of longer consultations when those doctors were bad at consulting anyway. The journal in which this article was published carried a dissenting letter (of which I was a co-author) making this point which is not cited. I accept that Lemon et al (2014) supports their view.

- There are a whole raft of research studies showing benefit for patients from longer consultations in general practice, some of which are cited in this article (Nos 14 and 18). There is also a systematic review (Wilson and Childs, 2006) demonstrating benefit for patients on several parameters. One reference they do cite, but not in this context, is Orton et al.(2016, I declare an interest), which showed that longer GP consultations are significantly more patient centered.

- Means of large numbers can conceal as well as reveal.

“Primary care”

‘Primary care’ has become an ambiguous term. It certainly includes nurse- led walk-in centres, pharmacy prescribing, probably ambulance outreach, and sometimes hospital accident and emergency services as well.

None of these are the same as general practices which the authors studied. It would be helpful if the authors clarified this.

RECOMMENDATIONS

That the authors be asked to report on the scatter between general practices in the recorded average duration of consultations by both GPs and nurses.

In particular, they should report the number and proportions of general practices in which the usual duration of consultations are:

< 5 minutes

5-7.49 minutes

7.5-9.9 minutes

10-11.9 minutes

12-14.9 minutes

15 minutes or more

They should review the sentence on page 9, para 1: “that consultation duration is not higher or lower across different ... general practices” in the light of these new analyses.

Without reporting on the scatter, they cannot make this claim, which may not be true.

RECOMMENDATION ON PUBLICATION

If they do these additional analyses and make appropriate adjustments to the text, then I recommend publication in BMJ Open.

Professor Sir Denis Pereira Gray OBE HonDSc FRCP FRCGP
FMedSci

Emeritus Professor, University of Exeter;. Consultant, St Leonards’
Research Practice, Exeter

Reference

Irving G and Holden J 15 minute consultations: better for patients
and GPs BMJ
2012; 344:31

REVIEWER	Matthew Howell NHS Greater Glasgow & Clyde Scotland
REVIEW RETURNED	20-Jul-2017

GENERAL COMMENTS	Interesting study. Useful results. Abstract: the results section could be more informative regarding subgroup analysis. The information given does not satisfactorily answer the question set by the aim in the abstract. "In 964,148 GP consultations duration was associated with patient gender, age, deprivation, and practice training status, and practice consultation rate" does not provide any real evidence and does not inform the reader whether this is a positive or negative association. It would be more useful to state which factors did influence consultation duration.
---

REVIEWER	Dr Farnaza Ariffin Universiti Teknologi MARA Selangor, Malaysia
REVIEW RETURNED	23-Jul-2017

GENERAL COMMENTS	Overall: This is an important study and the information are useful for Improvement of patient care within a general practice setting. Abstract: In the results section, suggest to omit the total number of GP consultation and nurse consultation as it may confuse readers at that point. Suggest to say "In GP consultation, duration was associated with..." Introduction: Adequate, no further comments nor suggestions. Methods:  1. Suggest to add a sentence to explain what is CPRD for non-UK audience e.g. CPRD is a research service that provides primary care records for the purpose of public health research (to include a reference link to CPRD) 2. For deprivation (quintile of IMD) suggest to include the definitions of how deprivation is categorized. Results: Suggest to define the type of random sampling conducted e.g. was it simple random? Discussion: The statement; "Despite these associations, the observed difference in duration were small...". Even though it may seem small but the results show significance in some categories therefore suggest instead to query whether these significance difference have any major impact or differences in patient care? In conclusion, an interesting and enjoyable read.
---

REVIEWER	Parker Magin University of Newcastle Australia
REVIEW RETURNED	30-Jul-2017

GENERAL COMMENTS	This is a well-conceived and well-conducted analysis. The principal findings – of consultation duration – will be of considerable importance in health systems planning, especially as both nurse and phone consultations are included in the findings. The associations of consultation duration are very interesting but interpretation is subject to a number of caveats related to potentially important factors in consultation duration. The authors rightly acknowledge the limitations in the range of independent variables for which they had data. Most important among these is lack of data on clinician characteristics and patient morbidities. Nevertheless, the associations found in the analyses are relevant and have potentially important implications. The conclusion is drawn from the findings that ‘patients are treated similarly regardless of background’ and ‘consultation duration is equitable’. This is fair enough, but interpretation could be a bit more nuanced. The authors’ interpretation that the slightly shorter duration of GP consultation for the lowest compared to highest quintile (5 seconds) is not clinically significant is very reasonable. But, given social determinants of health and relationship of deprivation to burden of disease, it could be argued that this finding does in fact represent patients not being treated similarly (if being treated similarly includes a consideration of clinical need – this is analogous to the argument that the authors mount regarding longer consultation times for older patients; ‘Implications’, paragraph 2). The difference in GP consultation duration between training and non-training practices has, as the authors suggest, potential implications for future GP training programs. Forty-four seconds difference per consultation, when multiplied across a full day’s or week’s consultations, may be a fairly modest but possibly clinically significant effect. For GP consultations, is there data on how much of this difference is contributed by trainee consultations and how much reflects consultations of trainers and other GPs in the practice? From the authors’ noting a lack of clinician characteristics data, I suspect that this can’t be explored. Thus this study reports overall data of considerable importance on consultation duration. Some associations of consultation duration established here are of intrinsic interest. The study also points the way to areas where further data is required to better understand the context and implications of these findings. Minor points  • ‘research standard’ data (Methods; paragraph 1) could be defined. • Article Summary, first dot point: ‘data known to be representative ‘ rather than ‘data known to representative ‘
---

VERSION 1 – AUTHOR RESPONSE

Reviewer: 1

Reviewer Name: Denis Pereira Gray

Dear Editor

PATIENT AND PRACTICE LEVEL FACTORS ASSOCIATED WITH CONSULTATION DURATION: A CROSS SECTIONAL ANALYSIS OF OVER ONE MILLION CONSULTATIONS IN ENGLISH PRIMARY CARE

Thank you for inviting me to assess this article which has been submitted for publication. I am pleased to do so.

I know several of the authors, including three of the professors, but I have no professional or financial links with any of the authors. I do not consider that I have any conflict of interest in commenting for you.

STRENGTHS

Comment: This article has many strengths:

- It has been written by a strong academic team from a leading British Department of General Practice/Primary Care.
- More than that, it comes from the School of Primary Care which links leading university departments of this kind.
- The authors have access to one of the biggest and best known general practice data bases in the UK or indeed in Europe.
- The methods used are appropriate for the data.
- Some of the results reported are new.
- One new finding is that for every 10% increase in the consultation rate, consultation duration decreased by 3 seconds, although the authors do not discuss if this is clinically important.

RESPONSE: We thank the reviewer for his detailed comments and are pleased that they agree that there are many strengths to our work. We have now added text to the discussion to comment on the clinical importance of our finding regarding consultation rates.

“We observed a small decrease of three seconds in the duration of GP consultations for every 10% increase in the rate of consultation, which is unlikely to be clinically important. However, increases in consultation rates above 10% could negatively impact clinical care.”

Comment: When I started reading I had a keen sense of interest and expectation, but when I had finished reading, I felt disappointed. This article is worthy but dull. The authors have not really thought enough about the potential of their data and have simply run standard analyses in standard ways. They can do better than this.

RESPONSE: We think it is unfortunate that the reviewer found some aspects of our paper “dull”, despite our findings being valid and original, with other reviewers finding the paper “enjoyable”. We are unsure what the reviewer feels could have been done better, other than the specific concerns listed below, which we have now addressed.

DISAPPOINTMENTS

Comment: Some findings previously published

- Many of their findings are already known. Indeed three of the authors (Hobbs et al ., 2016) have already reported (in The Lancet) the mean length of GP consultations at 9.4 minutes.

RESPONSE: We include this important baseline information in the text, to inform readers who may not be familiar with our previous work.

Comment: It has long been known that consultations with female patients are longer, although these authors show the gap is smaller than others only 8 seconds

RESPONSE: We would argue that our analysis with respect to patient gender is a particular novelty of our work, having shown that gender is differentially associated with GP/ nurse consultation duration, which to the best of our knowledge, has not been demonstrated previously. Rather than using 'standard analyses in standard ways', we have used multi-level modelling techniques to more accurately describe this association taking account of clustering effects at patient and practice levels and adjusting for a wide range of confounding factors. Previous studies used less sophisticated methods of analysis and rarely had such detailed information about confounders. Furthermore, as noted in our Introduction, previous studies may not provide reliable, contemporary estimates of association for the UK setting, having been conducted in other countries or many years ago.

Comment: Range/dispersion of average duration of consultations between practices

- In an article entitled duration of consultations one would expect a critical analysis of the variation in duration of consultations between different general practices in which patients on average get longer or shorter consultations.
- The whole pattern of NHS general practice since 1948 has been a steady lengthening of consultations. Although 10 minutes is the commonest unit of time, there have been calls for 15-minute booking (Irving et al., 2012)
- In the NHS general practice from which I am writing this review, I conducted routine surgery appointments booked at a duration of 12 minutes (five to the hour), 20 years ago. This Practice has provided consultations of average duration of 15-16 minutes for several years and there are other NHS practices in Exeter offering 15-minute appointments routinely.
- Longer consultations, particularly 15-minute consultations are now provided routinely in an important minority of general practices. The absence of any quantification of this important development is disappointing. Leading-edge general practice is particularly important and this article ignores it.
- I think these authors, with their massive database, should report on the range between general practices how many practices consult at an average of more than 10 minutes, what number and what percentage? This is particularly important for general practitioners or general practices providing an average of 15 or more minutes.
- The authors should, as they do in the article, report on the characteristics of such practices by training status, geography, practitioner characteristics etc.
- Similarly, the authors should report on how many general practices offer shorter than average ie 7.5 minutes or of 5-minute appointments and the associated characteristics of these practices.

RESPONSE: We thank the reviewer for his helpful suggestion. We have now summarized the average consultation duration in each practice and the characteristics of practices offering consultations of different length in Figure S1, Table S1, and Table S4. Since duration of consultation is only reported in CPRD to the nearest whole minute, we have grouped practices according to the following durations: <5, ≥5 and <8, ≥8 and <10, ≥10 and <12, ≥12 and <15 and ≥15 minutes. We have edited text in the methods and results section to reflect these additional analyses.

"Mean consultation duration across practices was examined using histograms. Practices were grouped according to their average consultation duration (<5, ≥5 and <8, ≥8 and <10, ≥10 and <12, ≥12 and <15 and ≥15 minutes) and differences in their characteristics described.

"A minority of practices conducted substantially shorter or longer consultations on average (Figure S1)."

“Practice characteristics by average length of GP consultation, are described in Table S1. Practices conducting longer consultations had a lower rate of GP consultation but the relationship between other characteristics was less clear.”

“Practice characteristics by average length of nurse consultation, are described in Table S4. Similarly to GP consultations, practices conducting longer consultations had a lower rate of nurse consultation.”

Comment: Extreme durations

- It is stated that “extreme durations” of all consultations studied which the authors dismiss (page 9) numbered 8.6% but this is more than one in 12 and so on average one per average surgery session.
- This seems high and is higher than figures found in general practices which monitor this. Some more comment would be helpful.

RESPONSE: We have expanded the discussion text to more clearly state the possible reasons for these extreme consultations and have conducted some additional sensitivity analyses in this respect. “Consultation duration in CPRD reflects the length of time a patient record is open within the practice computer system, recorded in whole minutes. There were instances in the data of very long (>60 minutes, 0.3%) and very short (apparent 0 minutes, 8.3%) consultations which we rounded to 60 minutes and 0.5 minutes respectively. Long consultations may occur for genuine clinical need, but also if a staff member forgets to close a record. Apparent short consultations may occur if a record is opened incorrectly, if details of a straightforward consultation are entered only at the end of a consultation, or if the type of consultation (e.g. administrative) was miscoded. However, average durations were in line with a standard 10-minute appointment window, and final model estimates were similar when excluding these extreme durations or including them without rounding (data not shown).”

Comment: Patient satisfaction

- On page 10 it is stated “that there is no strong evidence that consultation length is associated with patient satisfaction” and two supporting references (16, 17) are given. This is a rather dogmatic statement, which I personally do not believe to be true. One of these references (Elmore et al., 2016) in my opinion is invalid. This was study of GPs selected as having poor communication skills, so it is not possible to draw conclusions about the effect of longer consultations when those doctors were bad at consulting anyway. The journal in which this article was published carried a dissenting letter (of which I was a co-author) making this point which is not cited. I accept that Lemon et al (2014) supports their view.
- There are a whole raft of research studies showing benefit for patients from longer consultations in general practice, some of which are cited in this article (Nos 14 and 18). There is also a systematic review (Wilson and Childs, 2006) demonstrating benefit for patients on several parameters. One reference they do cite, but not in this context, is Orton et al. (2016, I declare an interest), which showed that longer GP consultations are significantly more patient centered.

RESPONSE: We accept the point the author makes regarding the cited study (Elmore et al.) and have edited this section to reflect the limitations of the study. We do, however, maintain our view that there is little strong evidence that increased consultation duration is associated with greater patient satisfaction. Indeed, the review cited by the reviewer (Wilson, 2006) reinforces this point and we have already cited this review as part of wider discussion of the benefits of longer consultations.

“However there is little strong evidence that consultation length is associated with patient satisfaction generally,[17,18] or when GPs are pre-selected for poor communication,[19] although one study has shown an association with more patient centeredness.[6]”

Comment: Means of large numbers can conceal as well as reveal.

RESPONSE: As with any analysis involving summary estimates, our work may not reveal particularly nuanced/ importantly different associations in certain subgroups of practices or patients. Such an investigation was not part of our pre-specified analysis plan and would therefore only be exploratory in nature. We believe that the large number of consultations studied is a strength of this work in comparison with previous literature.

Comment: "Primary care"

'Primary care' has become an ambiguous term. It certainly includes nurse- led walk-in centres, pharmacy prescribing, probably ambulance outreach, and sometimes hospital accident and emergency services as well. None of these are the same as general practices which the authors studied. It would be helpful if the authors clarified this.

RESPONSE: We believe that the methods and results sections are clear in stating that our analysis concern only those consultations occurring in general practice. We have, however, added a sentence to the strengths and limitation section to highlight possible areas in primary care where our results may not be generalizable.

"A limitation is the consideration of consultations occurring in general practice only and our results may not be generalizable to other settings (e.g. walk in centres)."

RECOMMENDATIONS

That the authors be asked to report on the scatter between general practices in the recorded average duration of consultations by both GPs and nurses. In particular, they should report the number and proportions of general practices in which the usual duration of consultations are:

< 5 minutes

5-7.49 minutes

7.5-9.9 minutes

10-11.9 minutes

12-14.9 minutes

15 minutes or more

They should review the sentence on page 9, para 1: "that consultation duration is not higher or lower across different ... general practices" in the light of these new analyses. Without reporting on the scatter, they cannot make this claim, which may not be true.

RESPONSE: As requested, we have now carried out these additional analyses (see text above). We have also edited the sentence in the first paragraph of the discussion to address the new findings. "Although there is some variation in mean duration across practices, this is not explained by many of the practice characteristics studied."

RECOMMENDATION ON PUBLICATION

If they do these additional analyses and make appropriate adjustments to the text, then I recommend publication in BMJ Open.

Professor Sir Denis Pereira Gray OBE HonDSc FRCP FRCGP FMedSci

Emeritus Professor, University of Exeter;. Consultant, St Leonards' Research Practice, Exeter

Reference

Irving G and Holden J 15 minute consultations: better for patients and GPs BMJ 2012; 344:31

Reviewer: 2

Reviewer Name: Matthew Howell

Comment: Interesting study. Useful results.

Abstract: the results section could be more informative regarding subgroup analysis. The information given does not satisfactorily answer the question set by the aim in the abstract. "In 964,148 GP consultations duration was associated with patient gender, age, deprivation, and practice training status, and practice consultation rate" does not provide any real evidence and does not inform the reader whether this is a positive or negative association. It would be more useful to state which factors did influence consultation duration.

RESPONSE: We thank the reviewer for pointing out this lack of detail. We have now amended the abstract to include information about the direction of observed associations.

"Results: Mean duration of face-to-face GP consultations was 9.24 minutes compared to 5.32 minutes for telephone consultations. Nurse face-to-face and telephone consultations lasted 9.70 and 5.73 minutes on average, respectively. Longer GP consultation duration was associated with female patient gender, practice training status and older patient age. Shorter duration was associated with higher deprivation and consultation rate. Longer nurse consultation duration was associated with male patient gender, older patient age and ever smoking; and shorter duration with higher consultation rate. Observed differences in duration were small (e.g. GP consultations with female patients compared to male patients were 8 seconds longer on average)."

Reviewer: 3

Reviewer Name: Dr Farnaza Ariffin

Overall: This is an important study and the information are useful for Improvement of patient care within a general practice setting.

Abstract: In the results section, suggest to omit the total number of GP consultation and nurse consultation as it may confuse readers at that point. Suggest to say "In GP consultation, duration was associated with..."

Introduction: Adequate, no further comments nor suggestions.

RESPONSE: We thank the reviewer for this suggestion and have modified the results section of the abstract accordingly.

"Results: Mean duration of face-to-face GP consultations was 9.24 minutes compared to 5.32 minutes for telephone consultations. Nurse face-to-face and telephone consultations lasted 9.70 and 5.73 minutes on average, respectively. Longer GP consultation duration was associated with female patient gender, practice training status and older patient age. Shorter duration was associated with higher deprivation and consultation rate. Longer nurse consultation duration was associated with male patient gender, older patient age and ever smoking; and shorter duration with higher consultation rate. Observed differences in duration were small (e.g. GP consultations with female patients compared to male patients were 8 seconds longer on average)."

Methods:

1. Suggest to add a sentence to explain what is CPRD for non-UK audience e.g. CPRD is a research service that provides primary care records for the purpose of public health research (to include a reference link to CPRD)

RESPONSE: We have now expanded this sentence to further explain what CPRD is, with reference to a recently published data resource profile.

"Consultation and patient data were obtained from the Clinical Practice Research Datalink (CPRD), a research database of anonymised patient records drawn from over 600 UK general practices.[9]"

2. For deprivation (quintile of IMD) suggest to include the definitions of how deprivation is categorized.

RESPONSE: IMD data was supplied directly by CPRD and was not calculated by ourselves. We have added text to the methods section to indicate this.

“IMD data was supplied in quintiles by CPRD, who link patient postcodes to publically available IMD scores and group data into quintiles at the English national level.”

Results: Suggest to define the type of random sampling conducted e.g. was it simple random?

RESPONSE: We have modified the text in the methods section to make explicit the use of simple random sampling.

“Due to data volume, analysis was limited to a 10% simple random sample from each age-sex strata of eligible patents and those who consulted at least once during the study period.”

Discussion: The statement; "Despite these associations, the observed difference in duration were small...". Even though it may seem small but the results show significance in some categories therefore suggest instead to query whether these significance difference have any major impact or differences in patient care?

RESPONSE: We have now added further text discussing possible clinical implications of our findings. “However, our findings that more deprived patients have shorter consultations on average could indicate inequalities based on clinical need since more deprived patients have higher rates of premature mortality.[15]”

“We observed a small decrease of three seconds in the duration of GP consultations for every 10% increase in the rate of consultation, which is unlikely to be clinically important. However, increases in consultation rates above 10% could negatively impact clinical care.”

In conclusion, an interesting and enjoyable read.

Reviewer: 4

Reviewer Name: Parker Magin

Comment: This is a well-conceived and well-conducted analysis.

The principal findings – of consultation duration – will be of considerable importance in health systems planning, especially as both nurse and phone consultations are included in the findings.

The associations of consultation duration are very interesting but interpretation is subject to a number of caveats related to potentially important factors in consultation duration. The authors rightly acknowledge the limitations in the range of independent variables for which they had data. Most important among these is lack of data on clinician characteristics and patient morbidities.

Nevertheless, the associations found in the analyses are relevant and have potentially important implications.

The conclusion is drawn from the findings that ‘patients are treated similarly regardless of background’ and ‘consultation duration is equitable’. This is fair enough, but interpretation could be a bit more nuanced. The authors’ interpretation that the slightly shorter duration of GP consultation for the lowest compared to highest quintile (5 seconds) is not clinically significant is very reasonable. But, given social determinants of health and relationship of deprivation to burden of disease, it could be argued that this finding does in fact represent patients not being treated similarly (if being treated similarly includes a consideration of clinical need – this is analogous to the argument that the authors mount regarding longer consultation times for older patients; ‘Implications’, paragraph 2).

RESPONSE: We thank the reviewer for highlighting this important consideration. We have now edited the Implications section to give a more balanced interpretation.

“We observed small absolute differences in consultation duration, despite statistical significance for some factors. This may suggest that all patients are treated similarly and that consultation duration is equitable and in line with patient need. For example we observed large differences related to patient age, which is likely to be confounded with comorbidity and complexity of consultation. However, our findings that more deprived patients have shorter consultations on average could indicate inequalities based on clinical need since more deprived patients have higher rates of premature mortality.[15]”

Comment: The difference in GP consultation duration between training and non-training practices has, as the authors suggest, potential implications for future GP training programs. Forty-four seconds difference per consultation, when multiplied across a full day's or week's consultations, may be a fairly modest but possibly clinically significant effect. For GP consultations, is there data on how much of this difference is contributed by trainee consultations and how much reflects consultations of trainers and other GPs in the practice? From the authors' noting a lack of clinician characteristics data, I suspect that this can't be explored.

RESPONSE: We thank the reviewer for his helpful comment. We have been able to conduct a sensitivity analysis investigating the association with consultations conducted with GPs who have a staff role indicating they are a “GP registrar”. For this we added an indicator variable into our final GP consultation model to indicate whether the GP entering the data into the record was a GP registrar or not. We found that consultations were on average 4.08 (95% CI 4.02 to 4.14) minutes longer with a GP registrar than otherwise. Furthermore, the indicator variable for whether the practice was a training practice or not became non-significant ($p=0.0656$). This may suggest that much of the original association with training practice status is driven by consultations conducted by GP registrars on their own, assuming that if a GP registrar was entering information into the patient record, it is unlikely that a more senior GP was present. However, we have no way of testing this assumption and do not possess more detailed data regarding other staff members present during the consultation. We also found that in those practices which were not identified as training practices in the national data, 4.9% of GP consultations appeared to be conducted by GP registrars (compared to 11.6% in training practices) so it is unclear to what extent the GP registrar staff role is used accurately. We have added text to the results section and discussion to reflect this additional analysis.

“In post-hoc sensitivity analysis, we explored whether the association of duration with practice training status may be driven by consultations with trainee GPs alone, by adding a variable into the final model to indicate whether the GP conducting the consultation was a registrar or not. We found that consultations were on average 245 seconds longer with a GP registrar than otherwise and practice training status became non-significant ($p=0.0656$, Table S3).”

“Our primary/ post-hoc results regarding the association of duration with practice training status/ GP registrars are consistent with the UK research. However, we found that in practices which were not identified as training practices in the national data, 4.9% of GP consultations appeared to be conducted by GP registrars (compared to 11.6% in training practices). This indicates inaccuracies in coding either of staff role or of training practice status, and hence this finding needs further replication in future studies.”

Comment: Thus this study reports overall data of considerable importance on consultation duration. Some associations of consultation duration established here are of intrinsic interest. The study also points the way to areas where further data is required to better understand the context and implications of these findings.

Minor points

- ‘research standard’ data (Methods; paragraph 1) could be defined.

RESPONSE: We have added more detail to this sentence to clarify this point.

“...were defined as “up-to-standard” (CPRD definition of continuous high quality data recording fit for use in research).

Comment: Article Summary, first dot point: 'data known to be representative ' rather than 'data known to representative '

RESPONSE: Thank you. We have corrected this omission accordingly

VERSION 2 – REVIEW

REVIEWER	PROFESSOR SIR DENIS PEREIRA GRAY Emeritus Professor University of Exeter; Consultant St Leonard's Research Practice Exeter
REVIEW RETURNED	08-Sep-2017

GENERAL COMMENTS	I find it interesting that in this study on 316 general practices only 1 was found providing consultations of 15 minutes or more, whilst there are three general practices providing this in my own health district. It has made me more cautious about a sample of only 218,304 patients being used to understand UK general practice nationally. Of course, I accept that their methods are clearly defined and that given this they are fully entitled report what they finding. I expect the single most important finding in this article will in time be judged to be the association between longer consultation time and lower consultation rate, which the authors report in their summary the other way round. However, by including this finding in a sentence which begins: "Longer nurse consultations.....with a semicolon leading into "and shorter duration with higher consultation rate." They make it clear this is true for nurses (page 8 last sentence of text). There is a logic in expressing this relationship starting with the duration of consultations placed first as this is a variable Managing GP Partners can and do alter, whereas the consultation rate comes later from searches. As I understand their figures, the important relationship is broader than just in relation to nurses and applies to GPs as well (line 7 in Table S1 implies this but they have calculated it precisely (page 7 for 10% increase in [GP] consultation rate, 1000 per 10,000 person years. GP consultations decreased by 3 seconds. If the authors agree that this finding applies to GPs, it would be logical to state this in the abstract too. They state that 3 seconds is unlikely to be clinically significant and I agree. However their new figures show variations in the mean duration of consultations of up to 50%. Variation in duration of consultations between general practices The authors underestimate the importance of their findings on the mean duration of consultations with GPs. Patients in the 30.4% of general practices who receive 10 or more minutes on average in GP consultations are at a considerable advantage over the 26.3% who receive 8 or fewer minutes on average per consultation. Given the strong evidence that longer GP consultations are more patient-centred, which they cite, and provide other advantages for patients too.
---

	In the abstract the adjective 'small' in relation to : "Observed differences in consultation duration.....are treated similarly regardless of background." reflects their previous data and analyses but is somewhat strong in relation to the new data that they provide. A two-minute increase on a consultation of 8 minutes (to 10 minutes or more) is of course 25%. Taking the arithmetic further and considering a typical nuclear family of two adults and two children. Hobbs et al (2016) show that each member of such a family will see a GP face-to-face three times a year on average each year. So the family will on average have 12 such GP consultations pa. Now, comparing the 30.4% of families in the >5 and < 8 minute average consultation group with 26.3% families in the > 10 <12 minute consultation duration group, then the latter families will receive at least on average $12 \times 2 = 24$ minutes more GP care per year, equivalent to two and a half extra GP consultations pa. I hope that the authors will review their wording in the summary. Given that, this is an important article which should now be published.
--	--

REVIEWER	Parker Magin Conjoint Professor School of Medicine and Public Health University of Newcastle Australia
REVIEW RETURNED	26-Aug-2017
GENERAL COMMENTS	The authors have satisfactorily addressed my comments and queries.

VERSION 2 – AUTHOR RESPONSE

Reviewer 1:

Thank you for sending the authors' response to my assessments with another copy of the revised article.

Comment: When I received the revised version, for some reason Table S1 did not print out , so I was under the impression that the authors had not responded to my original suggestion. Now, having seen Table S1 I am pleased to confirm that they have reported the additional analyses, which are important.

I find it interesting that in this study on 316 general practices only 1 was found providing consultations of 15 minutes or more, whilst there are three general practices providing this in my own health district. It has made me more cautious about a sample of only 218,304 patients being used to understand UK general practice nationally. Of course, I accept that their methods are clearly defined and that given this they are fully entitled report what they finding.

I expect the single most important finding in this article will in time be judged to be the association between longer consultation time and lower consultation rate, which the authors report in their summary the other way round. However, by including this finding in a sentence which begins: "Longer nurse consultations.....with a semicolon leading into "and shorter duration with higher consultation rate." They make it clear this is true for nurses (page 8 last sentence of text).

There is a logic in expressing this relationship starting with the duration of consultations placed first as this is a variable Managing GP Partners can and do alter, whereas the consultation rate comes later from searches. As I understand their figures, the important relationship is broader than just in relation to nurses and applies to GPs as well (line 7 in Table S1 implies this but they have calculated it precisely (page 7 for 10% increase in [GP] consultation rate, 1000 per 10,000 person years GP consultations decreased by 3 seconds)). If the authors agree that this finding applies to GPs, it would be logical to state this in the abstract too. They state that 3 seconds is unlikely to be clinically significant and I agree. However their new figures show variations in the mean duration of consultations of up to 50%.

RESPONSE: The author is correct in his assessment that the relationship between consultation duration and consultation rate is present for both GP and nurse consultations and we have already commented on this in the abstract as per the below text:

“Longer GP consultation duration was associated with female patient gender, practice training status and older patient age. Shorter duration was associated with higher deprivation and consultation rate.”

Comment: Variation in duration of consultations between general practices

The authors underestimate the importance of their findings on the mean duration of consultations with GPs. Patients in the 30.4% of general practices who receive 10 or more minutes on average in GP consultations are at a considerable advantage over the 26.3% who receive 8 or fewer minutes on average per consultation. Given the strong evidence that longer GP consultations are more patient-centred, which they cite, and provide other advantages for patients too.

In the abstract the adjective ‘small’ in relation to: “Observed differences in consultation duration.....are treated similarly regardless of background.” reflects their previous data and analyses but is somewhat strong in relation to the new data that they provide. A two-minute increase on a consultation of 8 minutes (to 10 minutes or more) is of course 25%. Taking the arithmetic further and considering a typical nuclear family of two adults and two children. Hobbs et al (2016) show that each member of such a family will see a GP face-to-face three times a year on average each year. So the family will on average have 12 such GP consultations pa. Now, comparing the 30.4% of families in the >5 and < 8 minute average consultation group with 26.3% families in the > 10 <12 minute consultation duration group, then the latter families will receive at least on average $12 \times 2 = 24$ minutes more GP care per year, equivalent to two and a half extra GP consultations pa.

I hope that the authors will review their wording in the summary.

RESPONSE: We thank the reviewer for his comments. Our abstract statement (“Small observed differences in consultation duration indicate that patients are treated similarly regardless of background”) refers to patient characteristics, rather than practice consultation rate, and we believe this statement is still valid. We have, however, edited our discussion section to comment further on the potential clinical significance of the observed association between consultation duration and rate as requested.

“Patients in the 31.7% of practices offering consultations less than eight minutes long may receive significantly less GP care compared to those in the 31% of practices providing consultations of 10 or more minutes long, particularly when considering this difference across multiple appointments. However, we also observed a decrease in duration of three seconds for every 10% increase in consultation rate indicating a degree of trade-off between consultation length and number.” We have also made a number of minor edits to the discussion (whilst retaining meaning), to accommodate the revised text and remain within manuscript word limits.

Reviewer: 4

The authors have satisfactorily addressed my comments and queries.

RESPONSE: Thank you. We are pleased our revisions our satisfactory.